# Comprehensive Analysis of Traffic Accidents in Seoul: Major Factors and Types Affecting Injury Severity

Hyeonchoel Jeong [1], Inhi Kim [2], Keejun Han [3,*] and Jungeun Kim [1,*]

1 Department of Computer Science and Engineering, Kongju National University, Cheonan 31080, Korea; hyunchul1357@naver.com

2 Department of Urban Systems Engineering, Kongju National University, Cheonan 31080, Korea; inhi.kim@kongju.ac.kr

3 Intelligent Convergence Research Laboratory, Electronics and Telecommunications Research Institute, Daejeon 34129, Korea

* Correspondence: keejun@etri.re.kr (K.H.); jekim@kongju.ac.kr (J.K.)

**Abstract:** Accident and fatality rates of traffic accidents worldwide are steadily increasing every year; thus, considerable effort has been made to prevent traffic accidents and prepare countermeasures. This study aims to identify the major factors and types that affect the severity of traffic accidents in Seoul by utilizing the Seoul Metropolitan Government's traffic accident dataset. To achieve this, we perform a comprehensive analysis by adopting various machine learning techniques—not only supervised learning methods but also unsupervised learning methods. As a result of the experiment, we derived several critical factors that were found to affect the severity of traffic accidents via supervised learning methods (i.e., ensemble-based and regression-based algorithms) and discovered dominant accident types via unsupervised learning methods (i.e., clustering-based algorithms). One of our primary findings is that, in contrast to common sense, environmental factors such as weather, season, and day of the week do not significantly affect the severity of traffic accidents in Seoul. Moreover, all methods highlight the importance of pedestrian-related factors, implying that it is highly necessary to prepare more meticulous institutional measures for pedestrians to reduce the negative influence of serious traffic accidents in Seoul.

**Keywords:** traffic accidents analysis; machine learning; logistic regression; XGBoost; DBSCAN

## 1. Introduction

Traffic accidents have emerged as a serious social problem today, as the number of car registrations has increased rapidly owing to global economic growth and improvements in living standards [1–3]. According to the report published by the World Health Organization (WHO) [4] in 2018, nearly 1.35 million people worldwide die in traffic accidents every year, implying that one person dies in a traffic accident every 24 s, an increase of 100,000 people compared to 2015. In addition, according to the Center for Disease Control and Prevention (CDC) [5], the cost of medical and productivity losses associated with deaths from car accidents in one year exceeds $63 billion. Therefore, it is necessary to identify major factors and types of traffic accidents to prevent traffic accidents in advance based on the results obtained.

Along these lines, a number of related studies and policies are being carried out abroad. However, there is still a lack of understanding of the major causes and mechanisms of serious traffic accidents in Seoul. Seoul is the largest city in South Korea, with various types of transportations used by almost 10 million citizens and vehicles every day, implying that the traffic accidents would cause tremendous social and economic losses.

The results of traffic accident data analysis may vary depending on the characteristics of the local traffic environment. Thus, it is necessary to focus on the intrinsic properties of Seoul for a deeper understanding of the causes and mechanisms of traffic accidents in

Seoul. Furthermore, traffic accidents are caused by a combination of various factors such as human-errors, road conditions, and environments. This means that we need to perform a comprehensive analysis of traffic accident datasets. Additionally, there is no single method that always yields the best results in all cases; therefore, various methodologies with different philosophies should be used for complex analysis.

In this study, we aim to identify the significant factors and types that affect the severity of traffic accidents by focusing on the cases of Seoul. To this end, we used big data on traffic accidents in Seoul pertaining to various factors by adopting three widely used machine learning techniques: ensemble-based, regression-based, and clustering-based methodologies. Throughout the analysis, we found that the severity of traffic accidents is mainly determined by pedestrian-related variables, not by driver-related variables, which is different from the results reported in previous studies [6–10]. We assume that this is because of the unique characteristics of Seoul, which has created a vehicle-oriented transportation environment that has been inevitably promoted by the daily traffic volume being so high, almost 10 million vehicles [11].

This paper makes the following contributions.

- We analyzed a set of features that affect the number of traffic accidents by classifying the features into three main factors—human, road, and environment—with a focus on Seoul, the capital of Republic of Korea.
- We unveiled the significant features that affect the severity of traffic accidents by exploiting various machine learning approaches: ensemble, regression, and clustering-based analytics.
- By performing further qualitative analysis, we suggest that establishing more preventive measures against pedestrian accidents would be an adequate approach to reduce the number of fatal injuries due to traffic accidents in Seoul.

The remainder of this paper is organized as follows: Section 2 reviews previous research. Section 3 describes the characteristics of the dataset we used for the analysis. Section 4 introduces three methods that we adopted, and Section 5 shows our findings based on the analysis using these methods. Finally, Section 6 concludes the study.

## 2. Related Work

Extensive studies have been carried out for the analysis of traffic accidents abroad. Chong et al. [12] used the traffic accident dataset provided by the General Estimates System (GES) to classify no injury, possible injury, non-incapacitating injury, incapacitating injury, and fatal injury classes for the period ranging from 1995 to 2000. The accuracy was compared and analyzed using decision trees, support vector machines, neural networks, and hybrid decision trees. The hybrid approach outperformed the other methodologies in the non-incapacitating injection, incapacitating injection, and fat injection classes, whereas decision trees have been shown to be the most suitable for classes with no injury or possible injury. Feng et al. [13] used the British traffic accident dataset to perform the a priori algorithm and then explored the rules for high support and lift. They found strong correlations in environmental characteristics, speed limits, and locations. Juan et al. [14] used Bayesian networks to analyze traffic accidents according to injury severity. Accident type, driver age, lighting, and the number of injuries were the most important factors associated with serious or fatal traffic accidents.

Zhao et al. [15] used the Bayesian network collision severity model to further analyze the complex combination relationship between single -and multi-vehicle traffic accidents. In addition, they ranked five factor combination sequences for the number of deaths and three-factor combination sequences for the number of injuries according to severity, thereby revealing the critical reason. Dong et al. [16] used a mixed logit model to investigate the difference in probability of accidents between single -and multi-vehicle accidents and used disaggregated data with response variables classified as no accidents, single-vehicle accidents, and multi-vehicle accidents. The analysis revealed that speed intervals, section lengths, and wet road surfaces are important for both single- and multi-vehicles, while most

other variables are important only for multi-vehicles. Thanapong et al. [17] conducted a study to determine ways to reduce rear and fatal rear collisions. To this end, a classification and regression tree (CART) was used, and the predictors for at-fault and not-at-fault driver models showed that the driver age was the most important, followed by the number of lanes and median opening area. Furthermore, the use of safety equipment was found to be the most important factor affecting fatality. Ahmed [18] applied a logistic regression to identify important variables affecting on traffic accident deaths. As a result of the experiment, the author showed that the major factors affecting traffic accident deaths were speed, location, and vehicle type.

Several research efforts have been devoted to the analysis of traffic accidents in Korea. Bhin and Son [19] investigated gender-related variables using the decision tree model to determine the severity of traffic accidents according to the gender of bus drivers and analyzed gender severity using the ordinal logit model. The analysis found that the signal violation variables of the violation of the law were commonly adopted by all genders, and that the same variables were adopted in the overall bus driver severity model and the male bus driver severity model, indicating that the carelessness of the driver greatly affected the severity of the accident. Lim et al. [20] analyzed traffic accident factors on roads with a width of less than 9 m using logistic regression models and found that drivers were driving straight and that women and pedestrians were driving bicycles. Kim et al. [21] conducted a study on the effects of traffic accidents on their occurrence according to the age of drivers, considering their human characteristics. Poisson regression analysis was used to develop a severity model for elderly and non-elderly people, and it showed that elderly drivers had an impact on their ability to predict stopping distance, discriminate surrounding situations, and respond to attention.

Table 1 summarizes related work. Compared with previous related studies, the major differences in this paper are as follows. First, we focused on Seoul, the capital of Republic of Korea. Since the results of traffic accident analysis can be different depending on the characteristics of the local traffic environment, it is crucial to consider the intrinsic properties of the target region. Second, we analyzed common trends and differences through both supervised and unsupervised learning methods. Finally, we reported new interesting findings that have not been reported previously.

**Table 1.** Summary of related work.

| Author | Objective | Analytical Algorithms |
|---|---|---|
| Chong et al. [12] | Performance comparison of four models to predict traffic accident severity | Decision tree, SVM, neural network, and hybrid decision tree |
| Feng et al. [13] | Discovery of dominant rules for traffic accidents | Apriori algorithm |
| Juan et al. [14] | Discovery of major factors affecting traffic accident severity | Bayesian network |
| Zhao et al. [15] | Discovery of major factors for traffic accidents | Bayesian network |
| Dong et al. [16] | Study of the difference in accident probability between single and multi-vehicle accidents | Mixed logit model |
| Thanapong et al. [17] | Discovery of factors that reduce rear-end collisions and fatal rear-end collisions | CART |
| Ahmed [18] | Identification of important variables for traffic accident deaths | Logistic regression |
| Bhin and Son [19] | Investigation of factors affecting the severity of accidents according to gender | Decision tree, ordered logit |
| Lim et al. [20] | Discovery of variables affecting traffic accidents on roads with a width of less than 9 m | Logistic regression |
| Kim et al. [21] | Investigation of driver skills by age and the influence of personality factors on the occurrence of traffic accidents | Poisson regression |

### 3. Characteristics of Dataset

The dataset we used includes 362,298 cases of traffic accidents that occurred between 2010 to 2018 in Seoul, provided by a public data portal [22]. The characteristics of the dataset are summarized in Table 2.

**Table 2.** Characteristics of the dataset.

| Factors | Variables | Data Type | Attribute Values |
| --- | --- | --- | --- |
| Human Factor | Accident type | Categorical | Side collision, backup collision, head-on collision, rear-end collision, crossing, passing on driveway, passing on the edge of the road, passing on the sidewalk, other |
| | Violation of law | Categorical | Non-compliance with safe driving obligation, not keeping a safe distance, signal violation, intersection crossing procedure violation, centerline violation, pedestrian protection obligation violation, speeding, other |
| | Perpetrator's gender | Categorical | Male, female, unidentified |
| | Perpetrator's age | Categorical | 1~117 years old, unidentified |
| | Victim's gender | Categorical | Male, female, unidentified |
| | Victim's age | Categorical | 1~117 years old, unidentified |
| Road Factor | Road surface | Categorical | Dry, wet, frozen, snow, flooding, unidentified |
| | Road type | Categorical | At an intersection, near an intersection, on a crosswalk, near a crosswalk, over the bridge, single road, crosswalk at an intersection, inside a tunnel, over an overpass, in an underpass, railroad crossing, unidentified, other |
| | Perpetrator's vehicle type | Categorical | Passenger car, lorry, two-wheeler, van, bicycle, motorized bicycle, heavy equipment, specialty vehicle, unidentified, other |
| | Victim's vehicle type | Categorical | Passenger car, lorry, two-wheeler, van, bicycle, motorized bicycle, heavy equipment, specialty vehicle, pedestrian, unidentified, other |
| Environmental Factor | Occurrence day | Categorical | 1 January 2010~31 December 2018 |
| | Occurrence time | Categorical | 00:00~23:00 |
| | Day of the week | Categorical | Monday, Tuesday, Wednesday, Thursday, Friday, Saturday, Sunday |
| | Weather | Categorical | Sunny, rainy, cloudy, snowy, foggy, unidentified |

In the pre-processing step, data with unknown values (i.e., null) were removed. Additionally, extremely low-frequency attributes (i.e., less than 0.01%) were removed because it is challenging to develop a good model if the data distribution is imbalanced.

The classification criteria for slight injury and serious injury are as follows: "slight injury" implies an injury that requires treatment for more than five days but less than three weeks due to a traffic accident. In contrast, "serious injury" implies an injury that requires treatment for at least three weeks due to a traffic accident. Further, "death" is considered as death within 30 days from the time of a traffic accident. In this study, serious accidents and deaths were equally treated as serious accidents, including life-threatening cases [23].

Because the data are all categorical data, we used one-hot encoding to transform them into a vector space model to use conventional machine learning algorithms. However, the use of one-hot encoding can dramatically increase the number of variables, resulting in poor classification performance of the algorithm. Therefore, we grouped the attribute values to reduce the number of attributes. For example, days of occurrence were grouped by season, and accident occurrence times were grouped into dawn (0: 00–6:00), day (6:00–18:00), and night (18:00–24:00). Further, the ages were grouped into underage (0–18 years), youth (19–34 years), middle-aged (35–49 years), old-aged (50–64 years), and elderly ($\geq$65 years). After pre-processing, the classification criteria and distribution ratios for each variable were

presented with a table separated by human factors, road factors, and environmental factors. A vast number of variables could be easily seen (Tables 3–5).

**Table 3.** Categories and frequencies of variables (human factor).

| Category | | Frequency | Ratio % | |
| --- | --- | --- | --- | --- |
| | | | Serious | Slight |
| Accident type | Side collision | 93,841 | 32.0 | 68.0 |
| | Backup collision | 574 | 11.8 | 88.2 |
| | Head-on collision | 10,444 | 41.9 | 58.1 |
| | Rear-end collision | 66,062 | 25.3 | 74.7 |
| | Crossing | 33,684 | 58.0 | 42.0 |
| | Passing on driveway | 7261 | 41.7 | 58.3 |
| | Passing on the edge of the road | 5399 | 35.6 | 64.4 |
| | Passing on the sidewalk | 3998 | 46.5 | 53.5 |
| | Other | 77,542 | 35.2 | 64.8 |
| Violation of law | Non-compliance with safe driving obligation | 158,966 | 35.0 | 65.0 |
| | Not keeping a safe distance | 44,245 | 24.5 | 75.5 |
| | Signal violation | 39,416 | 34.6 | 65.4 |
| | Pedestrian protection obligation violation | 12,384 | 29.6 | 70.4 |
| | Centerline violation | 11,297 | 43.5 | 56.5 |
| | Violation of pedestrian protection obligation | 10,754 | 51.7 | 48.3 |
| | Speeding | 455 | 78.7 | 21.3 |
| | Other | 21,288 | 32.2 | 67.8 |
| Perpetrator's gender | Male | 254,924 | 35.4 | 64.6 |
| | Female | 43,881 | 34.7 | 65.3 |
| Perpetrator's age | Underage | 10,424 | 37.4 | 62.6 |
| | Youth | 62,835 | 35.5 | 64.5 |
| | Middle-aged | 106,285 | 36.1 | 63.9 |
| | Old-aged | 89,769 | 34.4 | 65.6 |
| | Elderly | 29,492 | 34.7 | 65.3 |
| Victim's gender | Male | 224,221 | 33.2 | 66.8 |
| | Female | 74,584 | 41.7 | 58.3 |
| Victim's age | Underage | 20,436 | 37.3 | 62.7 |
| | Youth | 70,421 | 31.4 | 68.6 |
| | Middle-aged | 88,565 | 31.5 | 68.5 |
| | Old-aged | 89,212 | 36.0 | 64.0 |
| | Elderly | 30,171 | 52.1 | 47.9 |

**Table 4.** Categories and frequencies of variables (road factor).

| Category | | Frequency | Ratio % | |
| --- | --- | --- | --- | --- |
| | | | Serious | Slight |
| Road surface | Dry | 264,085 | 35.1 | 64.9 |
| | Wet | 31,596 | 37.0 | 63.0 |
| | Frozen | 2041 | 31.5 | 68.5 |
| | Snow | 1083 | 33.4 | 66.6 |
| Road type | At an intersection | 70,541 | 37.1 | 62.9 |
| | Near an intersection | 50,830 | 31.9 | 68.1 |
| | On a crosswalk | 11,889 | 54.5 | 45.5 |
| | Near a crosswalk | 6037 | 47.5 | 52.5 |
| | Over the bridge | 2916 | 32.8 | 67.2 |
| | Single road | 151,963 | 33.7 | 66.3 |
| | Crosswalk at an intersection | 2508 | 41.7 | 58.3 |

**Table 4.** *Cont.*

| Category | | Frequency | Ratio % | |
|---|---|---|---|---|
| | | | **Serious** | **Slight** |
| | Inside the tunnel | 773 | 33.6 | 66.4 |
| | Over the overpass | 755 | 29.4 | 70.6 |
| | In the underpass | 593 | 32.9 | 67.1 |
| Perpetrator's vehicle type | Passenger car | 215,661 | 33.3 | 66.7 |
| | Lorry | 24,215 | 41.0 | 59.0 |
| | Two-wheeler | 19,822 | 39.3 | 60.7 |
| | Van | 19,772 | 42.2 | 57.8 |
| | Bicycle | 9037 | 37.8 | 62.2 |
| | Motorized bicycle | 8251 | 38.6 | 61.4 |
| | Heavy equipment | 1530 | 45.2 | 54.8 |
| | Specialty vehicles | 517 | 44.9 | 55.1 |
| Victim's vehicle type | Passenger car | 137,014 | 33.9 | 76.1 |
| | Lorry | 9781 | 37.1 | 72.9 |
| | Two-wheeler | 31,428 | 46.1 | 53.9 |
| | Van | 13,235 | 33.0 | 67.0 |
| | Bicycle | 14,368 | 41.8 | 58.2 |
| | Motorized bicycle | 12,696 | 46.3 | 53.7 |
| | Heavy equipment | 472 | 30.9 | 69.1 |
| | Specialty vehicles | 277 | 28.2 | 71.8 |
| | Pedestrian | 79,534 | 49.8 | 50.2 |

**Table 5.** Categories and frequencies of variables (environmental factor).

| Category | | Frequency | Ratio % | |
|---|---|---|---|---|
| | | | **Serious** | **Slight** |
| Season | Spring | 78,545 | 35.8 | 64.2 |
| | Summer | 75,948 | 35.0 | 65.0 |
| | Autumn | 75,905 | 35.6 | 64.4 |
| | Winter | 68,407 | 34.6 | 65.4 |
| Time | Day | 152,061 | 35.2 | 64.8 |
| | Night | 93,772 | 34.5 | 65.5 |
| | Dawn | 52,972 | 36.9 | 63.1 |
| Day of week | Monday | 40,767 | 35.6 | 64.4 |
| | Tuesday | 42,735 | 35.0 | 65.0 |
| | Wednesday | 43,732 | 35.6 | 64.4 |
| | Thursday | 43,672 | 35.7 | 64.3 |
| | Friday | 47,497 | 35.5 | 64.5 |
| | Saturday | 46,122 | 34.3 | 65.7 |
| | Sunday | 34,280 | 35.3 | 64.7 |
| Weather | Sunny | 258,349 | 35.0 | 65.0 |
| | Rainy | 22,294 | 36.8 | 63.2 |
| | Foggy | 16,084 | 37.8 | 62.2 |
| | Snowy | 2078 | 33.3 | 66.7 |

*3.1. Human Factors*

Table 3 shows categories and frequencies of human factors. Human factors are classified into six categories: accident type, violation of law, perpetrator's gender, perpetrator's age, victim's gender, and victim's age. First, in the category of accident type, we can observe that crossing has the most significant influence on the severity of traffic accidents when considering the frequency and ratio of serious injuries. Second, in the category of the violation of law, speeding shows the highest ratio of serious injuries at 78.7 % despite the low frequency. Third, the perpetrator's gender and age have little effect on the severity of

the traffic accident, and the victim's gender and age have a more significant impact on the severity of the traffic accident when they are women or older adults.

### 3.2. Road Factors

Table 4 shows categories and frequencies of road factors. Road factors are classified into four categories: road surfaces, road types, perpetrator's vehicle types, and victim's vehicle types. First, the condition of the road surface does not have a significant effect on the severity of traffic accidents. When the road type is a crosswalk, accidents with more serious injuries occur. Furthermore, in the category of perpetrator's vehicle type, the proportion of accidents with serious injury is high in the order of heavy equipment and specialty vehicles, while in the category of victim's vehicle type, the pedestrian shows the highest ratio of accidents with serious injury at 49.8%.

### 3.3. Environment Factors

Table 5 shows categories and frequencies of environmental factors. Environmental factors are classified into four categories: season, time, day of the week, and weather. Examining the day of the week, it appears that Sunday has fewer traffic accidents than other days of the week. Further, examining the time, day of the week, and weather, there is little difference in the ratio of serious injury accidents. In particular, in contrast to common sense, it is interesting that the proportion of serious injuries on snowy or rainy days is not higher than that on other days. Thus, based on observations, it seems that there is not much connection between environmental factors and the severity of traffic accidents compared to other factors.

## 4. Analytical Methods

Widely used analytical methodologies, including ensemble-based and regression-based classifications, were applied to investigate the interrelationship between a dependent variable (i.e., the severity of traffic accidents) and independent variables (i.e., human, road, and environmental factors). We also adopted clustering to group data to determine the nature of each group so that we can discover dominant patterns of severe traffic accidents. In this work, we used eXtreme Gradient Boosting (XGBoost) because of its robustness for overfitting, which is critical for the classification problem [24]; logistic regression because of its superiority for handling categorical data [25]; and DBSCAN because of the freedom of the number of clusters [26].

### 4.1. XGBoost

XGBoost [24] is an ensemble algorithm that combines multiple decision trees and is a boosting-based model that improves the overfitting problems, speed, and stability of existing tree-based models. XGBoost sequentially trains a decision tree on the training data, and the objective function of XGBoost is defined as follows.

$$Obj(\theta) = \sum_{i=1}^{m} l\left(y_i, \hat{y}_i^{(t)}\right) + \sum_{k=1}^{K} \Omega(f_k), \quad \theta = (f_1, f_2, \ldots, f_K) \tag{1}$$

Here, $i$ represents the $i$th sample in the dataset and $m$ represents the total number of dataset inserted into the $k$th tree while $K$ is the total number of trees. $y_i$ is the class label, while $\hat{y}_i$ is the predicted label. $l$ is the loss function and $\Omega$ is the regularization term.

XGBoost adopts an additive strategy to improve the value of the objective function by adding a new decision tree to the previous one at each iteration. When the $t$-tree is constructed, the predicted value $\hat{y}_i^{(t)}$ can be formulated as follows.

$$\hat{y}_i^{(t)} = \sum_{k=1}^{t-1} f_k(x_i) + f_t(x_i) = \hat{y}_i^{(t-1)} + f_t(x_i) \tag{2}$$

According to Equations (1) and (2), the objective function can be formulated as follows.

$$Obj(\theta)^t = \sum_{i=1}^{m} l\left(y_i, \hat{y}_i^{(t-1)} + f_t(x_i)\right) + \sum_{k=1}^{t} \Omega(f_k) \tag{3}$$

If the tree contains a total of $T$ leaf nodes, the index of each leaf node is defined as $j$ and the weight of the samples for each leaf node is $w_j$. Then, the regularization term $\Omega(f)$ is defined as follows.

$$\Omega(f) = \gamma T + \frac{1}{2}\lambda \sum_{j=1}^{T} w_j^2 \tag{4}$$

Here, $\gamma$ and $\lambda$ represent penalty factors.

*4.2. Logistic Regression*

Regression analysis is a statistical technique for predicting the value of dependent variables from independent variables by understanding the causal relationship between variables. It is used to analyze the relevance of dependent variables to independent variables. A typical multiple linear regression (MLR) formula is equivalent to Equation (5).

$$p_{MLR}(y_i|x_i) = v + w^T x_i, \quad i = 1, 2, \ldots, m \tag{5}$$

where $X = [x_1, \ldots, x_m]^T \in \mathbb{R}^{m \times n}$ is a set of training data and $Y = [y_1, \ldots, y_m] \in \mathbb{R}^m$ is a set of labels. $w \in \mathbb{R}^n$ are weighting values and $v$ represents the intercept. $p_{MLR}(y_i|x_i)$ is the predicted value of $y_i$ when the independent variable $x_i$ attains a certain value. A typical regression analysis can acquire any value depending on the independent variable; thus, the $p_{MLR}(y_i|x_i)$ value can extend to infinity. If the dependent variable is a binary categorical variable, linear regression does not properly represent the relationship between the independent and dependent variables.

Therefore, logistic regression (LR) [25] can be used instead of linear regression if the dependent variable is binary (i.e., $y_i \in \{-1, +1\}$). Using logistic regression, the value of the dependent variable can be represented as a value between zero and one. Expressing logistic regression as a formula is equivalent to Equation (6).

$$p_{LR}(y_i|x_i) = \frac{exp\left((v + w^T x_i)y_i\right)}{1 + exp((v + w^T x_i)y_i)} \tag{6}$$

When the independent variable $x_i$ acquires a certain value, the predicted value of $p_{LR}(y_i|x_i)$ has the concept of probability between 0 and 1.

The average logistic loss function is calculated from the negative log-likelihood of the logistic model with respect to all samples.

$$\ell_{avg}(w,v) = \frac{1}{m} \sum_{i=1}^{m} \log\left(1 + \exp\left(-y_i\left(w^T x_i + v\right)\right)\right) \tag{7}$$

The model parameters w and v are determined in the direction of minimizing the average logistic loss function by a maximum likelihood estimation.

$$minimize \ \ell_{avg}(w,v) \tag{8}$$

By adding weight-regulating terms, which is a standard technique for preventing overfitting, to the mean logistic loss function, we can limit the weights from increasing in value and improve the generalization performance of our models.

$$minimize \ \ell_{avg}(w,v) + R(C) \tag{9}$$

Here, $R(C)$ is the regularization function, which can have different forms depending on the regularization method. The $\ell_1$-regularized logistic regression problem is

$$minimize \ \ \ell_{avg}(w, v) + \frac{1}{C} \sum_{i=1}^{n} |w_i| \tag{10}$$

The $\ell_2$-regularized logistic regression problem is

$$minimize \ \ \ell_{avg}(w, v) + \frac{1}{C} \sum_{i=1}^{n} w_i^2 \tag{11}$$

where $C$ is a regularization parameter used to adjust the balance between the magnitude of the weight vector and the average logistic loss measured by the $\ell_1$-norm or $\ell_2$-norm.

### 4.3. DBSCAN

DBSCAN [26] is an unsupervised learning method that clusters data with similar characteristics, clustering dense parts of the data. $D$ is the user's database, and point $p$, $q \in D$ is a $d$-dimensional vector. Further, $N_{eps}(p) = [q \in D | dist(p, q) \leq Eps]$ is the set of points in the radius $Eps$ centered on point $p$. When a point $p$ satisfies the $p \in N_{eps}(q)$ while $p$ is part of a set of $q$ and $N_{eps}(p) \geq minPts$, point $q$ is defined as the core point, and point $p$ is directly density-reachable from point $q$. Thus, if there are more than $minPts$ points within the $Eps$ radius at point $p$, then point $q$ is classified as a core point. If a chain exists where $p_{i+1}$ from point $p$ to $q$ is directly density-reachable from $p_i$, then point $p$ is defined as density-reachable from point $q$. However, if a density-reachable point $o$ exists from points $p$ and $q$, it is defined as density-connected. When $C_i$ is considered a cluster within $D$, we define $noise = \{p \in D | \forall i : p \notin C_i\}$ as a noise point, which is a point that does not belong to any cluster [26].

## 5. Results

Experimental analysis was done through the Seoul Metropolitan Government's traffic accident dataset with the following focuses: (i) critical factors affecting the severity of traffic accidents (Section 5.1) and (ii) representative types of traffic accidents (Section 5.2). All experiments were performed on a PC with AMD Ryzen 7 2700X Eight-Core Processor 3.7 GHz CPU and 32 Gbyte RAM, running Windows 10. All algorithms were implemented in Python. In the data preprocessing step, we used one-hot encoding to handle the categorical data. For supervised learning methods, the ratio between a training set and test set is 75/25. All results in this section are statistically significant since the $p$-values are less than a typical significance level 0.01. The source code of all experiments is fully available at https://github.com/hyunchul1357/traffic-accident-analysis (accessed on 13 January 2022).

### 5.1. Factor Analysis through XGBoost and Logistic Regression

In XGBoost, there are three hyper-parameters to be first optimized to prevent overfitting and increase accuracy. Table 6 lists the results of the hyper-parameter optimization. We have observed that when the learning rate is 0.1, the depth of the tree is 3, and the number of weak learners is 200, it achieves the highest accuracy, which is 68.95 %. However, the difference in accuracy according to hyper-parameter changes is not large. In general, hyper-parameter tuning is performed to prevent overfitting or underfitting the model in order to find accurate trends in the dataset. The reason that hyper-parameter optimization does not dramatically change the results is that the dataset has a clear tendency.

**Table 6.** Effects of hyper-parameters of XGBoost.

| Parameters | | | Accuracy % |
|---|---|---|---|
| **Learning rate** | **Estimators** | **Max Depth** | |
| **0.1** | 100 | 3 | 68.76 |
| | | 5 | 68.69 |
| | | 7 | 68.19 |
| | **200** | **3** | **68.95** |
| | | 5 | 68.42 |
| | | 7 | 67.71 |
| 0.2 | 100 | 3 | 68.90 |
| | | 5 | 68.54 |
| | | 7 | 67.63 |
| | 200 | 3 | 68.76 |
| | | 5 | 67.64 |
| | | 7 | 66.58 |

Table 7 lists the top five independent variables after learning the XGBoost.

**Table 7.** Importance of independent variables.

| Independent Variables | F Score |
|---|---|
| Victim's vehicle type = Pedestrian | 53 |
| Victim's vehicle type = Passenger car | 48 |
| Perpetrator's vehicle type = Passenger car | 47 |
| Violation of law = Signal violation | 46 |
| Victim's vehicle type = Two-wheeler | 46 |

Comparing Table 7 with Tables 3–5 shows that the victim's vehicle type = pedestrian, violation of law = signal violation, and victim's vehicle type = two-wheeler are considered important variables in judging serious accidents, while victim's vehicle type = passenger car and perpetrator's vehicle type = passenger car are considered important variables in determining slight accidents. In particular, the victim's vehicle type = pedestrian was chosen as the most important variable in determining the severity of traffic accidents.

In logistic regression, we need to choose L1 or L2 regularization and optimize the C value, which adjusts the degree of the fitting. Table 8 shows the results of the hyper-parameter optimization. Based on the experimental results, the C value was set to be 1 with L1 regularization. As in the case of XGBoost, even in logistic regression, the difference in accuracy according to hyperparameter changes is not large. This again supports the clear trend of the dataset.

**Table 8.** Effects of hyper-parameters of Logistic Regression.

| Regularization | C | RMSLE | RMSE | MAE | Training Set Accuracy % | Test Set Accuracy % |
|---|---|---|---|---|---|---|
| **L1** | 0.001 | 0.3955 | 0.5706 | 73.1889 | 67.43 | 67.44 |
| | 0.01 | 0.3904 | 0.5631 | 63.7788 | 68.23 | 68.30 |
| | 0.1 | 0.3894 | 0.5618 | 62.4428 | 68.35 | 68.44 |
| | **1** | **0.3896** | **0.5618** | **62.3266** | **68.37** | **68.44** |
| | 10 | 0.3896 | 0.5618 | 62.3238 | 68.36 | 68.44 |
| L2 | 0.001 | 0.3909 | 0.5639 | 65.8057 | 68.09 | 68.20 |
| | 0.01 | 0.3896 | 0.5621 | 62.7888 | 68.34 | 68.40 |
| | 0.1 | 0.3896 | 0.5619 | 62.3579 | 68.36 | 68.43 |
| | 1 | 0.3896 | 0.5618 | 62.3266 | 68.36 | 68.44 |
| | 10 | 0.3896 | 0.5618 | 62.3238 | 68.36 | 68.44 |

Table 9 shows the top 10 regression coefficients. A higher value means a higher influence on the severity of traffic accidents. The result shows that whether the perpetrator is speeding has the most significant impact and that the victim's vehicle type has a significant impact on serious traffic accidents for the case of a motorized bicycle, pedestrian, bicycle, and elderly victim. Additionally, when the perpetrator's vehicle type is a two-wheeler, it has a high impact on the severity of traffic accidents, supporting the claim that motorcyclists are more likely to be seriously injured in a traffic crash than people in passenger cars. In fact, the death rate for two-wheelers has constantly increased in Seoul from 2010 to 2018 due to the increase in the number of single households and the need for delivery services, although the total rate of death by traffic accidents slowly decreased during the same period.

**Table 9.** Ten Variables leading to serious traffic accidents.

| Variables | Regression Coefficient Values |
|---|---|
| Violation of law = Speeding | 1.8490 |
| Perpetrator's vehicle type = Two-wheeler | 0.6885 |
| Victim's vehicle type = Motorized bicycle | 0.6717 |
| Victim's age = Elderly | 0.5820 |
| Victim's vehicle type = Pedestrian | 0.5254 |
| Victim's vehicle type = Bicycle | 0.5200 |
| Violation of law = Centerline violation | 0.4805 |
| Road type = Crossing | 0.4100 |
| Violation of law = Signal violation | 0.4005 |
| Perpetrator's vehicle type = Heavy equipment | 0.3730 |

It is worth noting that the perpetrator's violation of law = signal violation, victim's vehicle type = pedestrian, and victim's vehicle type = two-wheeler variables are derived by not only logistic regression but also XGBoost as critical variables affecting the severity of traffic accidents. This demonstrates the necessity to prepare countermeasures against the perpetrators' signal violations and accidents involving two-wheeled vehicles and pedestrians.

Table 10 shows the bottom 10 regression coefficients. A lower value means a higher influence on slight traffic accidents. The results show that the backup collision is most closely related to slight accidents, followed by victim's vehicle type = passenger car and accident type = passing on the edge of the road. In general, the rate of slight accidents appears to be high because the vehicle speed is not high when backing up or passing along the edge of the road. In addition, in Table 4, when the victim's vehicle type is a passenger car, many slight accidents occur, and a similar trend appears in the regression analysis result.

**Table 10.** Ten variables leading to slight traffic accidents.

| Variables | Regression Coefficient Values |
|---|---|
| Accident type = Backup collision | −1.0902 |
| Victim's vehicle type = Passenger car | −0.4670 |
| Accident type = Passing on the edge of the road | −0.4471 |
| Victim's age = Underage | −0.3909 |
| Road type = Crosswalk at an intersection | −0.3828 |
| Victim's vehicle type = Lorry | −0.2660 |
| Perpetrator's vehicle type = Passenger car | −0.2199 |
| Road type = Near an intersection | −0.2105 |
| Victim's age = Youth | −0.2057 |
| Victim's gender = Male | −0.1959 |

*5.2. Cluster Analysis through DBSCAN*

In DBSCAN, there are two input parameters: (i) eps and (ii) minPts. We determined hyper-parameters following the heuristic suggested by literature [26,27]. MinPts is set to

approximately twice the dimensionality, thus 30 for a 14-dimensional space. Then, the value of eps is estimated by plotting the distance to the (MinPts−1)th nearest neighbor for each of sampled points, sorted in descending order, and finding the distance to an "elbow" of the curve. As a result of DBSCAN, we derived three major clusters corresponding to 97.8 of the entire dataset.

Table 11 shows the results of arranging the modes for each variable in each cluster after clustering. Given the high proportion of variables within the accident type = side collision and road type = at an intersection, cluster 1 appears to be a cluster for side collision accidents within intersections. Further, cluster 2 appears to be a cluster for rear-end collision accidents, given that the proportion of accident type = rear-end collision is high. In addition, cluster 3 is considered to be a cluster for pedestrian accidents, given that the proportion of the victim's vehicle type = pedestrian is 100%. In cluster 3, violation of law = non-compliance with safe-driving obligation was 92.1%, which was much higher than that of the other two clusters. It can be inferred that non-compliance with safe-driving obligation leads to many pedestrian accidents. Overall, the clustering results show that environmental factors do not significantly influence traffic accidents given the distribution ratio. This supports the previous results, in which environmental factors such as weather do not significantly impact the occurrence of traffic accidents in Seoul. In addition, it is also interesting to note that in all clusters, the gender of the victim is predominantly male.

**Table 11.** Characteristics of three major clusters.

| Variables | Cluster 1 | Cluster 2 | Cluster 3 |
|---|---|---|---|
| Season | Spring (26.5 %) | Spring (26.9 %) | Summer (27.3 %) |
| Time | Day (47.5 %) | Day (52.0 %) | Day (56.6 %) |
| Day of the week | Friday (16.1 %) | Saturday (17.2 %) | Friday (16.7 %) |
| Accident type | Side collision (57.6 %) | Rear-end collision (46.3 %) | Other (53.0 %) |
| Violation of law | Non-compliance with safe-driving obligation (45.3 %) | Non-compliance with safe-driving obligation (59.8 %) | Non-compliance with safe-driving obligation (92.1 %) |
| Road surface | Dry (98.3 %) | Dry (96.2 %) | Dry (99.4 %) |
| Weather | Sunny (97.6 %) | Sunny (95.7 %) | Sunny (98.4 %) |
| Road type | At an intersection (61.1 %) | Single road (95.5 %) | Single road (95.0 %) |
| Perpetrator's vehicle type | Passenger car (87.9 %) | Passenger car (86.5 %) | Passenger car (77.2 %) |
| Perpetrator's gender | Male (90.4 %) | Male (89.9 %) | Male (84.9 %) |
| Perpetrator's age | Middle-aged (43.7 %) | Middle-aged (40.0 %) | Middle-aged (40.7 %) |
| Victim's vehicle type | Passenger car (80.3 %) | Passenger car (75.2 %) | Pedestrian (100 %) |
| Victim's gender | Male (89.1 %) | Male (85.9 %) | Male (54.2 %) |
| Victim's age | Middle-aged (38.9 %) | Old-aged (36.8 %) | Middle-aged (25.1 %) |

## 6. Conclusions and Discussion

In this study, we used a traffic accident dataset, which included accidents in Seoul from 2010 to 2018, to identify the major factors and types that affect the severity of traffic accidents. To create a good classification, less frequent or skewed data were pre-processed by being removed and re-grouped, and analyzed using XGBoost, Logistic Regression, and

DBSCAN, which are the representative methodologies widely used in the field. In the XGBoost results, the case where the perpetrator violated the signal or the victim was riding a two-wheeled vehicle was also found to be an important variable in judging a serious traffic accident. In addition, the case where the victim and perpetrator's vehicle type was a passenger car had a significant influence in judging the slight accident. In logistic regression, the top and bottom 10 variables were analyzed according to the regression coefficient values to identify factors affecting the severity of traffic accidents. As a result, the perpetrator's violation of the law was found to affect serious traffic accidents in the order of speeding, two-wheeler or motorized bicycle, elderly, pedestrian, and bicycle. In contrast, environmental factors did not significantly affect traffic accidents. The clustering analysis results derived the top three clusters, represented by in-intersection side-crashes, rear-end collision, and clusters for pedestrians. Considering the three methodologies as a whole, environmental factors such as season, day of the week, and weather were found to be insignificant on the severity of traffic accidents. On the other hand, it is worth noting that variables for pedestrians appear in common among all of the three approaches, which would suggest establishing more preventive measures against pedestrian accidents, in order to reduce the fatal injury by the traffic accidents in Seoul.

In practice, actual traffic accidents are caused by a combination of more specific and diverse factors than the variables in the dataset used in this study. For example, in this study, variables such as speeding in violation of the law may vary depending on how fast the analysis was conducted. Factors such as driver's vision or seat belt wearing may also affect the outcome. If data including more diverse information are available, a more specific analysis will be possible; thus, an active data opening policy is needed. Notwithstanding the aforementioned limitations, our study still provides important insights on the unique and important features related to traffic conditions in Seoul for furthering the city's traffic safety.

**Author Contributions:** Conceptualization, J.K. and K.H.; methodology, J.K., K.H., I.K. and H.J.; software, H.J.; validation, J.K., I.K. and H.J.; investigation, H.J.; resources, J.K. and H.J.; data curation, H.J.; writing—original draft preparation, J.K. and H.J.; writing—review and editing, J.K., K.H. and I.K.; visualization, J.K., K.H. and H.J.; supervision, J.K. and K.H.; project administration, J.K. and K.H.; funding acquisition, J.K. All authors have read and agreed to the published version of the manuscript.

**Funding:** This research was supported by the research grant of the Kongju National University in 2021 and by the National Research Foundation of Korea (NRF) grant funded by the Korea government (MSIT) (No. 2021R1A4A1031509).

**Institutional Review Board Statement:** Not applicable.

**Informed Consent Statement:** Not applicable.

**Data Availability Statement:** The data presented in this study are available at Public Data Portal, http://www.data.go.kr (accessed on 18 March 2021).

**Acknowledgments:** The author would like to extend their thanks to reviewers and editors for helping to improve this paper.

**Conflicts of Interest:** The authors declare no conflict of interest.

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
