# Peer review of "Comprehensive Analysis of Traffic Accidents in Seoul: Major Factors and Types Affecting Injury Severity"

_applsci, doi:10.3390/app12041790_

Round 1

Reviewer 1 Report

I am reiterating my previous high grade of the high value of this manuscript. The manuscript sent to me for review entitled "Comprehensive Analysis of Traffic Accidents in Seoul: Major Factors and Types Affecting Injury Severity" is a valuable contribution to science in terms of content, methodology, empirical, cognitive, and application. The authors resubmitted the manuscript, refined it in many ways, addressing the suggestions. They corrected the shortcomings, which only confirms my belief that the manuscript is valuable research material.

The authors added quite important fragments of the text from the methodological point of view.

I do not see any shortcomings, I even notice the strengths of the manuscript itself. Without wishing to repeat me, I will refer to the discussions and conclusions, which after the abstract are one of the fragments that determine whether to read the entire article or not. The mature discussion and conclusions deserve special attention. The authors focused on the methodological layer, which in my opinion is particularly interesting. They approached the pragmatic side of applied science. I read the manuscript again with great appreciation.

Congratulations and good luck!

Author Response

Reviewer 1

  1. I am reiterating my previous high grade of the high value of this manuscript. The manuscript sent to me for review entitled "Comprehensive Analysis of Traffic Accidents in Seoul: Major Factors and Types Affecting Injury Severity" is a valuable contribution to science in terms of content, methodology, empirical, cognitive, and application. The authors resubmitted the manuscript, refined it in many ways, addressing the suggestions. They corrected the shortcomings, which only confirms my belief that the manuscript is valuable research material.

Answer: Thanks for the generous comments.

Reviewer 2 Report

The paper discusses traffic accidents in Seoul through the usage of data analytics techniques to unveil its main factors.

The structure of the paper is good and presents an adequate list and analysis of related work in the area. However, the contributions are not clear.

Session 3 needs more details in terms of how the methods have been used in the dataset with justification as to why it is effective for the problem at hand.

To facilitate the discussion, the variables in table 6 could be renamed, to better map what has been described in table 3. In other words, the " _ " in the variable names doesn’t really help in the discussion section.

Table 5 demonstrates very little difference between the accuracies, statistically speaking there is very little support to the max depth as 3. It might need a deeper analysis/discussion here. A similar situation happens with table 7, the differences are not statistically significant to justify some of the decisions taken.

The paper contributions are not entirely clear. The paper claims that the contributions are:

1) We analyzed a set of features that affect traffic accidents by classifying the features into three main factors: human, road, and environment

However, is this an original contribution or would it build on previous work? You might want to discuss this deeper.

2) We unveiled the significant features that affect the severity of traffic accidents by exploiting various machine learning approaches: ensemble, regression, and clustering-based analytics

However, the analysis and the results seem to not statistically compare the differences between these features. It is not clear if the differences are statistically significant or not. I would recommend adding a statistical analysis to support the claims.

Also, tables need to be better organised and formatted. Finally, there are some spelling and, mainly, grammar mistakes that should be corrected

Author Response

Reviewer 2

  1. Session 3 needs more details in terms of how the methods have been used in the dataset with justification as to why it is effective for the problem at hand.

Answer: They are very good points. The main reason for using Logistic Regression is that the target label of the data is binary categorical data. Other regression algorithms only work well with continuous variables and are therefore not suitable for the data used in this study. Among ensemble-tree algorithms, XGBoost is robust for overfitting which is critical for the classification problem. Most clustering algorithms require the user to determine the number of clusters before doing clustering. However, since the types of traffic accidents are very diverse and complex, it is practically difficult to determine the number of clusters before clustering analysis. DBSCAN, one of the clustering algorithms, does not need to set the number of clusters in advance, and it finds clusters with an unspecified distribution well. We added these explanations in the paper. (p.8)

  1. To facilitate the discussion, the variables in table 6 could be renamed, to better map what has been described in table 3. In other words, the " _ " in the variable names doesn’t really help in the discussion section.

Answer: Thank you for a nice recommendation. Per your comment, we revised the variable name in Table 6, 8, and 9. Also, we made variable names italic for better readability. (pp. 11-13)

  1. Table 5 demonstrates very little difference between the accuracies, statistically speaking there is very little support to the max depth as 3. It might need a deeper analysis/discussion here. A similar situation happens with table 7, the differences are not statistically significant to justify some of the decisions taken.

Answer: You are right. Obviously, optimizing hyper-parameters of XGBoost and logistic regression is not sensitive in this dataset. In general, hyper-parameter tuning is performed to prevent overfitting or underfitting the model in order to find accurate trends in the data set. The reason that hyper- parameter tuning does not dramatically change the results is that the data set has a clear tendency. It does not mean that results are not statistically significant. The two are completely different. We added this explanation in the paper. (pp. 10-11)

  1. The paper contributions are not entirely clear. The paper claims that the contributions are:

1) We analyzed a set of features that affect traffic accidents by classifying the features into three main factors: human, road, and environment

However, is this an original contribution or would it build on previous work? You might want to discuss this deeper.

Answer: That’s a good point. The analysis of various features by subdividing them into three factors is our new contribution to a deeper understanding. However, more importantly, the analysis was attempted in consideration of the characteristics of the area by focusing on the Seoul area. We modified the main contributions in the paper. (p.2)

2) We unveiled the significant features that affect the severity of traffic accidents by exploiting various machine learning approaches: ensemble, regression, and clustering-based analytics

However, the analysis and the results seem to not statistically compare the differences between these features. It is not clear if the differences are statistically significant or not. I would recommend adding a statistical analysis to support the claims.

Answer: It is thought to be a comment connected to the 3rd comment. It is true that the accuracy of the model according to the change of the hyper-parameter value is similar. (Please see Tables 5 and 7) However, there is a clear difference in the importance of variables that affect accuracy. (Please see Tables 6, 8, and 9) These two are completely different. To avoid misunderstanding, we added this explanation in the paper. (pp. 10-11)

  1. Also, tables need to be better organised and formatted. Finally, there are some spelling and, mainly, grammar mistakes that should be corrected

Answer: They are good points. Per your comments, we reorganized tables and fixed grammar errors.

Reviewer 3 Report

In this paper, the authors aim to identify the significant factors and types that affect the severity of traffic accidents by focusing on the cases of Seoul.

Contributions made through this paper are as follows:

  • analyze a set of features that affect traffic accidents by classifying the features into three main factors: human, road, and environment;
  • exploiting various machine learning approaches for unveiling the significant features that affect the severity of traffic accidents;
  • suggestions preventive measures to reduce the fatal injury by traffic accidents in Seoul.

The authors have fulfilled everything they promised and stated in the introduction of the paper, but I think that small additions and corrections are needed in the article.

  • Given that the related work is presented at the end of the article, the correlation of the obtained indicators, which are the most common causes of traffic accidents, is expected, with the indicators obtained by researchers in the papers presented in the related work chapter. That is, it is necessary to explain how the obtained results are similar/identical to the results of the researchers listed in the chapter related work and how you explain the differences.
  • The upper bound of the age interval of participants in traffic accidents listed in Table 1 is 117 years. Is this really accurate information?

''Perpetrator's age Categorical 1 year old ~ 117 year old, and Victim's age Categorical 1 year old ~ 117 year old''

Line 150: ''Y can be expressed as shown in Equation 4 using Equations 1, 2, and 3.

? = ?(?) + ?(?) + ?(?) + ?(?)''

In expression 3 there is no label e (x), nor is it explained what the label e (x) means.

  • All labels used in formulas should be written in the same font in the text. (This is mainly related to Chapter 3.1.)

Author Response

Reviewer 3

  1. Given that the related work is presented at the end of the article, the correlation of the obtained indicators, which are the most common causes of traffic accidents, is expected, with the indicators obtained by researchers in the papers presented in the related work chapter. That is, it is necessary to explain how the obtained results are similar/identical to the results of the researchers listed in the chapter related work and how you explain the differences.

Answer: That’s a good point. Per your comment, we added a brief comparison of related studies and ours. (p.3)

  1. The upper bound of the age interval of participants in traffic accidents listed in Table 1 is 117 years. Is this really accurate information? ''Perpetrator's age Categorical 1 year old ~ 117 year old, and Victim's age Categorical 1 year old ~ 117 year old''

Answer: That’s a good point. It is true that both victims and perpetrators range in age from 1 to 117 years old according to the official guide [13] of this data set. The vehicle type of the 1-year-old perpetrator was ‘bicycle (1 case)’. (This is a minor accident in which a car collided with a car due to a signal violation at an intersection.) The vehicle type of the 117-year-old perpetrator was ‘a passenger car (1 case)’. In addition, when the victim was 1 year old, the vehicle types were ‘pedestrian (83 cases)’ and ‘bicycle (2 cases)’. When the victim is 117 years old, the vehicle type is 'Pedestrian (1 case)'.

  1. Line 150: ''Y can be expressed as shown in Equation 4 using Equations 1, 2, and 3.

? = ?(?) + ?(?) + ?(?) + ?(?)''

In expression 3 there is no label e(x), nor is it explained what the label e(x) means.

All labels used in formulas should be written in the same font in the text. (This is mainly related to Chapter 3.1.)

Answer: They are good points. Per your comment, we added the explanation for the label e(x). In addition, we revised all formulas to be more detailed. (pp. 8)

Reviewer 4 Report

Your paper deals with an important topic that can bring interesting result with potential to improve safety on road. I appreciate you have used more than just one method to analyse the dataset.

Regarding the formal aspects of your paper, consider restructuring your article – chapter 5 Related work would fit better in front of your own work –that is right after the introduction as chapter 2.

There are just few issues that are not clear and you should describe or clarify them in more detail:

 In table 1 in the list of Perpetrator’s vehicle type values I find strange Pedestrian is missing. Is it correct? Pedestrians definitely are the perpetrators of traffic accidents

More results of the DBSCAN cluster analysis in 4.2 are worth mentioning and explaining.  The 92,1% non-compliance with safe-driving obligation in Cluster 3 seems interesting. Furthermore perpetrator’s gender – male prevailing in all three clusters is worth some comment.

The list of recommendations in the conclusion is outside the scope of the paper and you should lay more emphasis on the fact that there are just the possibilities, but the particular measures require lots of analysis before application.

Author Response

Reviewer 4

  1. Regarding the formal aspects of your paper, consider restructuring your article – chapter 5 Related work would fit better in front of your own work –that is right after the introduction as chapter 2.

Answer: Thank you for a nice recommendation. Per your comment, we placed “Related Work” in Chapter 2.

  1. In table 1 in the list of Perpetrator’s vehicle type values I find strange Pedestrian is missing. Is it correct? Pedestrians definitely are the perpetrators of traffic accidents.

Answer: That’s a good point. We agree with you that pedestrians often would be a potential factor in certain traffic accidents. Unfortunately, however, in the dataset we used, there is no pedestrian value for perpetrators of traffic accidents.

  1. More results of the DBSCAN cluster analysis in 4.2 are worth mentioning and explaining. The 92.1% non-compliance with safe-driving obligation in Cluster 3 seems interesting. Furthermore perpetrator’s gender – male prevailing in all three clusters is worth some comment.

Answer: Thank you for a nice recommendation. Per your comment, we added more details of cluster analysis results including you mentioned. (p.13)

  1. The list of recommendations in the conclusion is outside the scope of the paper and you should lay more emphasis on the fact that there are just the possibilities, but the particular measures require lots of analysis before application.

Answer: They are good points. Per your comment, in the conclusions and discussion section, we deleted or toned down content that did not fit the topic of the paper. (p.14)

Round 2

Reviewer 2 Report

The authors have addressed all elements required for adjustment.

This manuscript is a resubmission of an earlier submission. The following is a list of the peer review reports and author responses from that submission.

Round 1

Reviewer 1 Report

The manuscript submitted to me for review entitled "Comprehensive Analysis of Traffic Accidents in Seoul: Major Factors and Types Affecting Injury Severity" is a valuable contribution to science in terms of content, methodology, empirical, cognitive and application.

Although the selected literature requires referring to newer items, and the manuscript contains more than 50% of items published more than 5 years ago, the whole is a current, universal, innovative contribution in terms of applied sciences. The study in the near future should also be extended to other areas and years.

The authors of the study were mature, showing the methodology, pointing to tools, and bearing in mind the knowledge that machine learning is unique in terms of its results. They shared the same workshop for recreating research steps, sharing knowledge and methodological insights.

The abstract contains all the necessary elements, it can be a separate product fully summarizing the manuscript. The keywords have been chosen correctly.

The introduction contains an introduction to the subject, the problem and goal are properly formulated, the methods used to achieve the goal were carefully selected. The introduction also briefly describes the organizational structure of the manuscript.

The study was conducted on the example of 362,298 observations, it relates to road accidents, the spatial scope covers 9 years from 2010-2018, the spatial scope: Seuoul. The study was performed on the basis of secondary data from public databases (they should be referenced under the tabular results and in other places where it is necessary).

The authors presented the study of factors in a group of factors using a factor matrix (qualitative method), specifying the type of the variable (categorical, qualitative, nominal). Only in the case of time data can they be quantified. The title in Table 1 is incorrect. It should read "Characteristics of the dataset".

Tables 2-4 show the frequencies for a given variable category, thus illustrating the distribution of the studied phenomena. A similar technique is used in the case of data mining to search for leading factors in the development of phenomena / processes (in the field of industrial statistics: data minig).

In section 3, the author presents the various stages of reaching the results precisely and transparently. They presented their approach using comprehensible mathematical equations that do not disturb the reception of the entire manuscript. The equations are presented clearly, simply and formally complete the whole.

The authors used the Python language for the empirical part, describing the environment in which they worked. In their research, they used 100 and 200 estimators at the learning threshold (0.1 and 0.2), using 3 levels of max deph: 3, 5, 7.

They used the F score to examine the significance of the variables, and they properly selected the measure for the scale of the variables. This means that they also have in-depth knowledge of statistics, in addition to IT knowledge.

The quality of the tested models is moderate, oscillating around the value of 67-68%. It is a predictable value, just like forecast errors for unpredictable phenomena, such as road accidents. The obtained results prove that the authors really reliably worked on real data, not manipulating the results within them.

Table 8-9 shows the regression coefficients that have been properly ordered. The authors consciously ordered the studied variables, indicating which factors have the greatest impact on the phenomenon under study (positive and negative impact).

Table 10 presents the characteristics of the 3 main clusters from the point of view of the multidimensional analysis of statistical features.

Although one could argue about the structure of the article, take a pattern: first a literature review, then a description of data and methods, then a presentation of results, and finally a discussion and conclusions. However, it was the authors who started from the reverse order (first they presented the methodology and data description, then the results, and then they referred to the literature review). Their approach is mature and conscious. They related the obtained research results to related works, straining out those that were not consistent in terms of the problem studied. Thus, they saved the reader from reading unrelated scientific achievements, bypassed the chaos, focusing only on the layer of research that is strongly convergent with their research. In this regard, they focused on the quality of the selected literature, not the quantity, which is undoubtedly positive. The emphasis was placed mainly on the analytical part, not the review part, which is justified from the point of view of the manuscript readability.

The mature discussion and conclusions deserve special attention. The authors focused on the methodological layer, which in my opinion is particularly interesting. They actually approached the pragmatic side of the applied science.

The manuscript is an interesting contribution to science, especially applied science, but also to practice. From the point of view of modeling and forecasting traffic, as well as designing infrastructure solutions (traffic management controllers), the results provide a lot of clues to think about. The work should be published because it brings a novelty to the methodological, empirical and cognitive gap, important from the point of view of road safety. It can be hoped that the subsequent research steps will allow the authors to formulate certain regularities or laws for the so-called "black swan" (I strongly recommend to read N.N. Taleb - Black Swan)

Side Notes:

Standardize fonts in references, e.g. reference 1.

No periods are used after table titles. There is no source or an indication that this is an author's work under the tables.

Congratulations and best regards,

Reviewer

Author Response

We deeply appreciate careful comments from the reviewers and thoroughly addressed all the comments. The updated or added sentences are colored in red in the draft. The following is the summary of our revision.

Reviewer 1

1. The study was performed on the basis of secondary data from public databases (they should be referenced under the tabular results and in other places where it is necessary).
Answer: Thank you for a nice recommendation. The reference to the data set is cited where the data set was first introduced. Also, we specified the reference to the data set in the Data Availability Statement. Additionally, we added the reference of data source under the tabular results. (p.3)

2. The title in Table 1 is incorrect. It should read "Characteristics of the dataset". 
Answer: Thank you for a nice recommendation. Per your comment, we revised the title of Table 1 from Statistics of the dataset to Characteristics of the dataset. (p.2)

3. Standardize fonts in references, e.g. reference 1.
Answer: That's a good point. The fonts of all references have been standardized. (pp. 15-16)

4. No periods are used after table titles. 
Answer: Thank you for a nice recommendation. We remove periods after table titles.

Reviewer 2 Report

This manuscript utilities to identify the major factors and types that affect the severity of traffic accidents in Seoul by utilizing the Seoul Metropolitan Government's traffic accident dataset. The paper is well written in general. It is recommended to address the following comments:

  1. The dataset used in this paper is all traffic accidents that occurred between 2010 to 2018 in Seoul. I consider it includes single vehicle, vehicle to vehicle and multi-vehicle accidents. While the attributes include perpetrator and victim. Hence, if the accident is a single vehicle crash, how to show the variables about perpetrator and victim?
  2. “ Data is between 2010 to 2018 in Seoul” (Page 2): Are there any year-specific factors affecting the crashes? Can you get the robust result from a single dataset where the nine-year data are simply pooled.
  3. For XGBoost, you choose the top 5 independent variables. In logistic regression, you analyze top 10 regression coefficients. Why you choose top 5 or top 10? What is the criteria?
  4. What are the proportion of training set and test set? Are the three methods chosen same proportions?
  5. This study apply three algorithms to investigate critical factors, the manuscript does not clearly describe why choose these algorithms. The objectives, scope, advantages, and disadvantages of the application of each method should be extensively presented.

Author Response

We deeply appreciate careful comments from the reviewers and thoroughly addressed all the comments. The updated or added sentences are colored in red in the draft. The following is the summary of our revision.

Reviewer 2

1. The dataset used in this paper is all traffic accidents that occurred between 2010 to 2018 in Seoul. I consider it includes single vehicle, vehicle to vehicle and multi-vehicle accidents. While the attributes include perpetrator and victim. Hence, if the accident is a single vehicle crash, how to show the variables about perpetrator and victim?
Answer: That's a good point. According to the official guide [13] of this data set, in the majority of single-vehicle accidents (falls, overturns, and departures from the road), vehicles are the perpetrators. However, parked vehicles were considered victims.

2. “Data is between 2010 to 2018 in Seoul” (Page 2): Are there any year-specific factors affecting the crashes? Can you get the robust result from a single dataset where the nine-year data are simply pooled.
Answer: That's a good point. Actually, the dataset used in this paper is marked with year information. We agreed that the results would be more interesting if we could find the year-specific factors affecting accidents, however, we could not find any. 

3. For XGBoost, you choose the top 5 independent variables. In logistic regression, you analyze top 10 regression coefficients. Why you choose top 5 or top 10? What is the criteria?
Answer: That's a good point. Determining k when deriving top k results is a very important and difficult problem. Thus, we consulted previous studies. According to the referenced paper [28, 24], k was chosen as 5 for decision tree-based methods and 10 for regression-based methods. 

4. What are the proportion of training set and test set? Are the three methods chosen same proportions?
Answer: That's a good point. The proportion of the training dataset and the test data set is a very important setting for conducting the experiment, and it is our fault for not specifying it in the paper. In our experiments, the proportion of the training dataset and the test dataset is 75:25. We added this explanation in the paper. (p.8)

5. This study apply three algorithms to investigate critical factors, the manuscript does not clearly describe why choose these algorithms. The objectives, scope, advantages, and disadvantages of the application of each method should be extensively presented.
Answer: They are very good points. The main reason for using Logistic Regression is that the target label of the data is binary categorical data. Other regression algorithms only work well with continuous variables and are therefore not suitable for the data used in this study. Among ensemble-tree algorithms, XGBoost is robust for overfitting which is critical for the classification problem. Most clustering algorithms require the user to determine the number of clusters before doing clustering. However, since the types of traffic accidents are very diverse and complex, it is practically difficult to determine the number of clusters before clustering analysis. DBSCAN, one of the clustering algorithms, does not need to set the number of clusters in advance, and it finds clusters with an unspecified distribution well. We added these explanations in the paper. (p.7)

Reviewer 3 Report

Overall, the paper is well written, but the novelty is limited in the algorithm aspect. This study aims to identify the major factors and types that affect the severity of traffic accidents in Seoul by utilizing the Seoul Metropolitan Government's traffic accident dataset. I believe the topic of this research is of interest to the traffic community

  1. Add a Table to summarize the related work
  2. “In the pre-processing step, data with unknown values were removed. Also, extremely low-frequency attributes were removed or re-grouped”, Describe the details on how to remove them since it is important for the subsequent analysis.
  3. Add reference for XGBoost, LR, and DBSCAN
  4. In the description of XGBoost, LR, and DBSCAN, explain the reason for choosing those algorithms, since there are many methods for classification, regression, and clustering.
  5. A summary of the contribution should be given.
  6. The discussion can be further improved.
  7. No public repository with the experiments carried out was mentioned. Making experiments available is a common practice in Machine Learning studies that facilitates the transparency and reproducibility of the results. I strongly recommend authors adopt this type of beneficial practice for the community.

Author Response

We deeply appreciate careful comments from the reviewers and thoroughly addressed all the comments. The updated or added sentences are colored in red in the draft. The following is the summary of our revision.

Reviewer 3

1. Add a Table to summarize the related work
Answer: Thank you for a nice recommendation. We added the table which summarizes related work. (pp. 13-14)

2. “In the pre-processing step, data with unknown values were removed. Also, extremely low-frequency attributes were removed or re-grouped”, Describe the details on how to remove them since it is important for the subsequent analysis.
Answer: They are good points. First, the unknown value means null or a value marked as “unknown” in the dataset itself. Second, extremely low-frequency attributes mean attributes whose frequency is in single digits or less than 0.01% of the total 362,298 data. It is not controversial to define extremely low-frequency attributes since the tendency of the data distribution are very clear. (Please refer to the dataset after preprocessing, https://github.com/hyunchul1357/traffic-accident-analysis). Also, detailed description of re-grouping is given on page 3 (Section 2). In order to clarify these issues, we revised the paper. (p. 2)

3. Add reference for XGBoost, LR, and DBSCAN
Answer: Thank you for a nice recommendation. We added the references for XGBoost, Logistic Regression, and DBSCAN. (p.7)

4. In the description of XGBoost, LR, and DBSCAN, explain the reason for choosing those algorithms, since there are many methods for classification, regression, and clustering.
Answer: They are very good points. The main reason for using Logistic Regression is that the target label of the data is binary categorical data. Other regression algorithms only work well with continuous variables and are therefore not suitable for the data used in this study. Among ensemble-tree algorithms, XGBoost is robust for overfitting which is critical for the classification problem. Most clustering algorithms require the user to determine the number of clusters before doing clustering. However, since the types of traffic accidents are very diverse and complex, it is practically difficult to determine the number of clusters before clustering analysis. DBSCAN, one of the clustering algorithms, does not need to set the number of clusters in advance, and it finds clusters with an unspecified distribution well. We added these explanations in the paper. (p.7)

5. A summary of the contribution should be given.
Answer: Thank you for a nice recommendation. Per your comment, we added the summary of contributions in the paper. (p.2)

6. The discussion can be further improved.
Answer: Thank you for a nice recommendation. Per your comment, we further improved the discussion. (pp. 14-15)

7. No public repository with the experiments carried out was mentioned. Making experiments available is a common practice in Machine Learning studies that facilitates the transparency and reproducibility of the results. I strongly recommend authors adopt this type of beneficial practice for the community.
Answer: Thank you for a nice recommendation. Per your comments, we have made the source code publicly available using GitHub (https://github.com/hyunchul1357/traffic-accident-analysis) and specify the URL in the paper. (p.8)

Round 2

Reviewer 2 Report

The authors have made some modifications according to the comments, and I suggest that this paper can be accepted.